# Scalable Bayesian inference of dendritic voltage via spatiotemporal recurrent state space models

**Ruoxi Sun**[*]
Columbia University

**Scott W. Linderman**[*]
Stanford University

**Ian August Kinsella**
Columbia University

**Liam Paninski**
Columbia University

## Abstract

Recent advances in optical voltage sensors have brought us closer to a critical goal in cellular neuroscience: imaging the full spatiotemporal voltage on a dendritic tree. However, current sensors and imaging approaches still face significant limitations in SNR and sampling frequency; therefore statistical denoising methods remain critical for understanding single-trial spatiotemporal dendritic voltage dynamics. Previous denoising approaches were either based on an inadequate linear voltage model or scaled poorly to large trees. Here we introduce a scalable fully Bayesian approach. We develop a generative nonlinear model that requires few parameters per dendritic compartment but is nonetheless flexible enough to sample realistic spatiotemporal data. The model captures potentially different dynamics in each compartment and leverages biophysical knowledge to constrain intra- and inter-compartmental dynamics. We obtain a full posterior distribution over spatiotemporal voltage via an efficient augmented block-Gibbs sampling algorithm. The nonlinear smoother model outperforms previously developed linear methods, and scales to much larger systems than previous methods based on sequential Monte Carlo approaches.

## 1 Introduction

Recent progress in the development of voltage indicators [1–8] has brought us closer to a long-standing goal in cellular neuroscience: imaging the full spatiotemporal voltage on a dendritic tree. These recordings have the potential (pun not intended) to resolve fundamental questions about the computations performed by dendrites — questions that have remained open for more than a century [9, 10]. Unfortunately, despite accelerating progress, currently available voltage indicators and imaging technologies provide data that is noisy and sparse in time and space. Our goal in this work is to take this noisy, sparse data and output Bayesian estimates, with uncertainty, of the spatiotemporal voltage on the tree, at arbitrary resolution.

A number of generic denoisers are available. For example, one previous approach is to run an independent spline smoother on the temporal trace from each pixel [1]. However, this approach ignores two critical features of the data. First, the data is highly spatiotemporally structured; thus, running a purely temporal smoother and ignoring spatial information (or vice versa) is suboptimal. Second, the smoothness of voltage data is highly inhomogeneous; for example, action potentials are much less smooth than are subthreshold voltage dynamics, and it is suboptimal to enforce the same level of smoothness in these two very different regimes.

How can we exploit our strong priors, based on decades of biophysics research, about the highly-structured dynamics governing voltage on the dendritic tree? Our starting point is the cable equation [11], the partial differential equation that specifies the evolution of membrane potential in space and time. If we divide up the tree into $N$ discrete compartments, then letting $V_t^{(n)}$ denote the voltage

---

[*]Equal contribution

of compartment $n$ at time $t$, we have

$$V_{t+\Delta t}^{(n)} \approx V_t^{(n)} + \frac{\Delta t}{C_n} \left[ \sum_j I_t^{(n,j)} + \sum_{n'=1}^{N} g_{nn'} \cdot (V_t^{(n')} - V_t^{(n)}) \right]. \tag{1}$$

Each compartment $n$ has its own membrane capacitance $C_n$ and internal currents $I_t^{(n,j)}$; $j$ indexes membrane channel types, with $j = 0$ denoting the current driven by the membrane leak conductance. The currents through each channel type for $j > 0$ in turn depend on the local voltage $V_t^{(n)}$ plus auxiliary channel state variables with nonlinear, voltage-dependent dynamics. The coupling of voltage and channel state variables renders the intra-compartment dynamics highly nonlinear.

Additional current flows between compartments $n$ and $n'$ according to the conductance $g_{nn'} \geq 0$ and the voltage drop $V_t^{(n')} - V_t^{(n)}$. The conductances are undirected so that $g_{nn'} = g_{n'n}$. The symmetric matrix of conductances $G = \{g_{nn'}\} \in \mathbb{R}^{N \times N}$ specifies a weighted, undirected tree graph. Nonzero entries indicate the strength of connection between two physically coupled compartments: if $g_{nn'}$ is large then voltage differences between compartments $n$ and $n'$ are resolved quickly, i.e. their voltages become more tightly coupled.

The Hodgkin-Huxley (HH) model [12] and its generalizations [13] offer biophysically detailed models of voltage dynamics, but learning and inference pose significant challenges in the resulting high-dimensional nonlinear dynamical system. Two approaches have been pursued in the past. First, we can restrict attention to the subthreshold regime, where the dynamics can be approximated as linear. Invoking the central limit theorem leads to a Gaussian approximation on the current noise (due to the sum over $j$ in Eq. 1), resulting in an overall linear-Gaussian model. If the observed data can in turn be modeled as a linear function of the voltage plus Gaussian noise, we are left with a classical Kalman filter model. Paninski [14] develops efficient methods to scale inference in this Kalman filter model to handle large trees. However, the resulting smoother doesn't handle spikes well — it either over-smooths spikes or under-smooths subthreshold voltages. The Kalman model suffers because it assumes voltage has one uniform smoothness level, and as already discussed, this assumption only makes sense in the subthreshold regime.

Alternatively, we can attempt to perform inference on noisy voltage recordings based on model (1) directly. There are many compartments, each with a voltage and a collection of channel state variables, leading to a very high-dimensional nonlinear dynamical system. For low-dimensional models, like single compartment models with few channel states, methods like sequential Monte Carlo (SMC) and approximate Bayesian computation (ABC) can be applied [15–17], but even with recent advances [18–20], inference in large scale biophysical models remains difficult. The learning problem (i.e. estimating the parameters governing the intra- and inter-compartment dynamics) is even harder: inaccurate state inferences lead to errors in parameter estimation, and poor parameter estimates lead to incorrect state inferences. Compounding all of this is model misspecification — critical parameters such as time constants and voltage-sensitivity functions vary across channels, and there are dozens of types of channels in real cells [13]—and model identifiability — many channel combinations can produce similar dynamics [21]. Thus performing inference on the multi-compartment biophysical model (1) directly seems intractable.

Below we propose an alternative approach, blending the cable equation model with general purpose statistical models of nonlinear dynamical systems, to enable efficient learning and inference of spatiotemporal voltage dynamics. Code is available at: https://github.com/SunRuoxi/Voltage_Smoothing_with_rSLDS.

## 2  Model

Our basic strategy is as follows. For each compartment $n$, the biophysical model in (1) involves potentially dozens of channel types, each with their own state variables evolving according to nonlinear dynamics — but to infer voltages (not individual currents) we only actually need the sum of the induced currents. We replace this sum with a simpler, low-dimensional effective model. We retain the basic spatial biophysical constraints on voltage dynamics (i.e., leaving the second term in model (1) as is), while approximating the nonlinear interactions between voltage and membrane channels with a more tractable model: a recurrent switching linear dynamical system (rSLDS). Figure 1 shows how each compartment is given its own discrete states, continuous states, and voltage, and how these variables interact to produce nonlinear spatiotemporal dynamics.

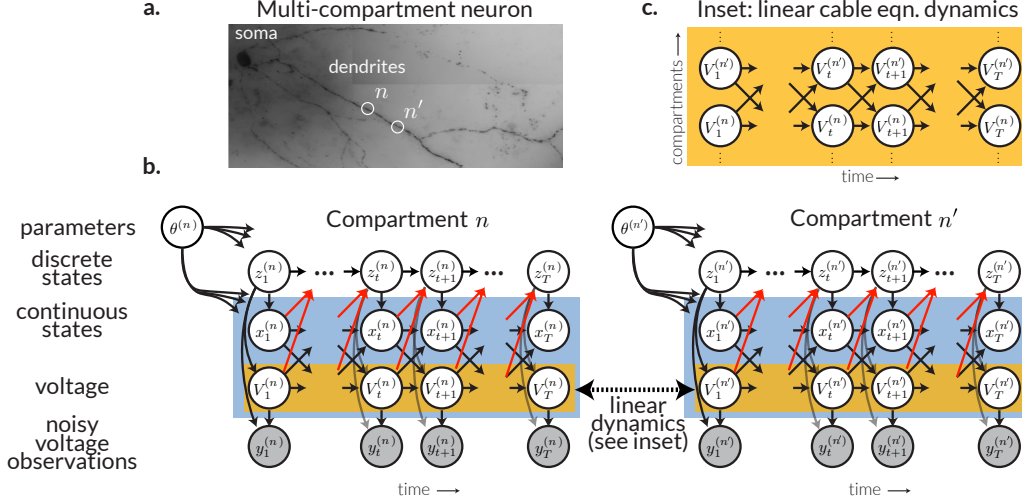

**Figure 1:** *Approximating the biophysical model of membrane potential dynamics with a tractable graphical model.* **a.** We study voltage dynamics on dendritic trees like the one shown here. **b.** Each compartment of the cell is approximated with a recurrent switching linear dynamical system, which has discrete latent states, continuous latent states, a true (unobserved) voltage, and noisy voltage observations. The continuous states and voltage follow piecewise linear dynamics conditioned on the discrete states; marginalizing over the discrete states, we obtain nonlinear dynamics within each compartment. The red arrows denote the *recurrent* dependencies by which continuous states and voltage modulate discrete transition probabilities. **c.** Importantly, the inter-compartmental voltage dependencies are linear, as specified by the cable equation.

For each compartment $n$, we introduce $x_t^{(n)} \in \mathbb{R}^D$, a continuous latent state of dimension $D$. (We set $D = 1$ in all the examples shown below, but higher-dimensional dynamics are possible). Let $z_t^{(n)} \in \{1, \ldots, K\}$ denote a corresponding discrete latent state; as we will see, these will correspond to different phases of the action potential. Given the discrete state, the voltage and continuous latent, $(V_t^{(n)}, x_t^{(n)})^\top$, together follow linear dynamics,

$$\mathbb{E}\left[ \begin{pmatrix} V_{t+\Delta t}^{(n)} \\ x_{t+\Delta t}^{(n)} \end{pmatrix} \,\middle|\, z_t^{(n)} = k, \left\{ V_t^{(n')} \right\}_{n' \neq n} \right]$$
$$= \begin{pmatrix} V_t^{(n)} \\ x_t^{(n)} \end{pmatrix} + \frac{\Delta t}{C_n} \left[ \underbrace{A_k^{(n)} \begin{pmatrix} V_t^{(n)} \\ x_t^{(n)} \end{pmatrix} + b_k^{(n)}}_{\approx \sum_j I_t^{(n,j)}} + \begin{pmatrix} \sum_{n'=1}^N g_{nn'} \cdot \left( V_t^{(n')} - V_t^{(n)} \right) \\ 0 \end{pmatrix} \right].$$

The dynamics matrix $A_k^{(n)} \in \mathbb{R}^{(D+1) \times (D+1)}$ and the bias vector $b_k^{(n)} \in \mathbb{R}^{D+1}$ parameterize linear dynamical systems for each discrete state in each compartment. We further assume additive Gaussian dynamics noise for each compartment and discrete state with covariance $Q_k^{(n)}$. Importantly, the voltage dynamics retain the inter-compartmental linear terms from the cable equation, linking connected compartments in the dendritic tree (inset of Fig. 1). On the other hand, the nonlinear summed intra-compartment currents $\sum_j I_t^{(n,j)}$ are replaced with a collection of piecewise linear dynamics on voltage and continuous states; by switching between these discrete linear dynamics, we can approximate the nonlinear dynamics of the original model (since any sufficiently smooth function can be approximated with a piecewise-linear function).

To complete the dynamics model, we must specify the dynamics of the discrete states $z_t^{(n)}$. We use a rSLDS, allowing the discrete states to depend on the preceding voltage and continuous states,

$$p(z_{t+\Delta t}^{(n)} = k \mid V_t^{(n)}, x_t^{(n)}) \propto \exp \left\{ w_k^{(n)\top} \begin{pmatrix} V_t^{(n)} \\ x_t^{(n)} \end{pmatrix} + d_k^{(n)} \right\}. \tag{2}$$

The red arrows in Fig. 1 highlight these dependencies. The linear hyperplanes defined by $\{w_k^{(n)}, d_k^{(n)}\}_{k=1}^K$ define a weak partition of the space of voltages and continuous latent states.

As the magnitude of the weight vectors increases, the partition becomes more and more deterministic. In the infinite limit, the discrete states are fully determined by the voltage and continuous states, and the switching linear dynamical system becomes a piecewise linear dynamical system [22]. We find that these models admit tractable learning and inference algorithms, and that they can provide a good approximation to the nonlinear dynamics of membrane potential.

Finally, we observe noisy samples of the voltage $y_t^{(n)} \sim \mathcal{N}(V_t^{(n)}, \sigma_k^{(n)2})$ for each compartment, with state-dependent noise. Our goal is to learn the parameters of the multi-compartment rSLDS, $\Theta = \{\{\theta^{(n)}\}_{n=1}^N, G\}$, where $\theta^{(n)} = \{A_k^{(n)}, b_k^{(n)}, Q_k^{(n)}, w_k^{(n)}, d_k^{(n)}, \sigma_k^{(n)}\}_{k=1}^K$. Given the learned parameters, we seek a Bayesian estimate of the voltage given the noisy observations.

## 3   Bayesian learning and inference

The recurrent switching linear dynamical system (rSLDS) inherits some of the computational advantages of the standard switching linear dynamical system (SLDS); namely, the conditional distribution of the discrete states given the continuous states is a chain-structured discrete graphical model, and most model parameters admit conjugate updates. However, the additional dependency in (2) breaks the linear Gaussian structure of the conditional distribution on voltage and continuous latent states.

Following previous work [23, 24], we use Pólya-gamma augmentation [25] to render the model conjugate and amenable to an efficient Gibbs sampling algorithm. Briefly, we introduce augmentation variables $\omega_{tk}^{(n)}$ for each compartment, discrete state, and time bin. After augmentation, the voltage and continuous states are rendered conditionally linear and Gaussian; we sample them from their complete conditional with standard message passing routines. Moreover, the augmentation variables are conditionally independent of one another and Pólya-gamma distributed; we appeal to fast methods [26] for sampling these. The discrete states retain their chain-structured conditionals, just as in a hidden Markov model. Finally, the model parameters all admit conjugate Gaussian or matrix-normal inverse Wishart prior distributions as in [23]; we sample from their complete conditionals.

One algorithmic choice remains: how to update across compartments? Note that all of the voltages and continuous latent states are jointly Gaussian given the discrete states and the augmentation variables. Moreover, within single compartments, the variables follow a Gaussian chain-structured model (i.e., a Kalman smoother model). We leverage this structure to develop a block-Gibbs sampling algorithm that jointly updates each compartment's voltage and continuous latent trajectories simultaneously, given the discrete states, augmentation variables, and the voltages of neighboring compartments[2].

## 4   Experimental Results

We evaluate the spatiotemporal recurrent SLDS and the corresponding inference algorithms on a variety of semi-synthetic datasets. First, we show that we can model real voltage recordings from a single compartment. Then, we evaluate performance on multi-compartment models, including a simulated dendritic branch and a real dendritic tree morphology.

### 4.1   A single-compartment rSLDS is a useful model of real somatic voltage

As a first test of the model and algorithm, we use intracellular voltage traces from the Allen Institute for Brain Science Cell Types Atlas [27, cell id=464212183]. The traces are recorded via patch clamp, a high signal-to-noise (SNR) method that provides "ground truth" measurements. We added artificial white noise with a standard deviation of 5mV to these recordings in order to test the model's ability to both learn membrane potential dynamics and smooth noisy data. We fit the rSLDS to 100ms of data sampled at $\Delta t = 0.1$ms, for a total of 1000 time points. We used three discrete latent states ($K = 3$) and one additional latent dimension ($D = 1$); already, this simple model is sufficient to provide a good model of the observed data. Figure 2(a) shows the observed voltage (i.e. the ground truth plus white noise), and the 95% posterior credible intervals estimated with samples from the posterior under the estimated rSLDS model. The inferred discrete state sequence (c) shows how the rSLDS segments the voltage trajectory into periods of roughly linear accumulation (blue), spike rise (yellow) and spike fall (red). These periods are each approximated with linear dynamics in $(V, x)$ space, as shown in

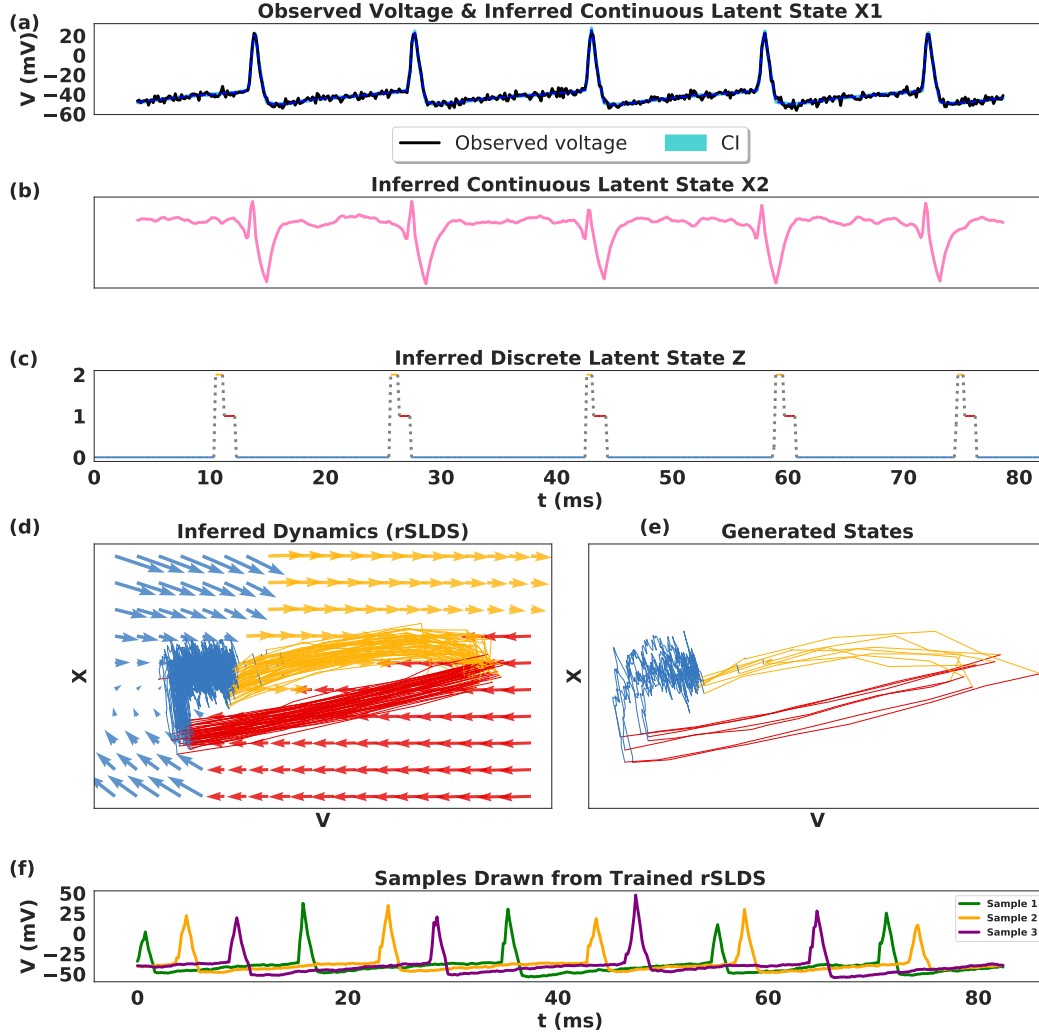

**Figure 2:** *A single-compartment, three-state rSLDS provides an adequate model of real somatic voltage responses.* (a) Observed voltage (formed by adding noise to the intracellular voltage from a real neuron) and the credible interval (CI) output by a the estimated rSLDS model. (b) Inferred continuous latent state $x(t)$ corresponding to this trial. (c) Inferred discrete latent state $z(t)$. (d) Inferred two-dimensional dynamics. The blue state (corresponding to $z = 0$ in (c)) is a fixed point at the rest potential; the yellow state corresponds to a fast depolarization (the upswing of the spike; $z = 2$) and the red state the hyperpolarization (the downswing of the spike; $z = 1$), followed by a return to the blue rest state. The thin traces indicate samples from $x(t)$ and $V(t)$ given the observed noisy voltage data. (e) Generative samples from the learned rSLDS; the difference between these traces and those shown in the previous panel is that these traces are generated using the learned rSLDS parameters without conditioning on the observed noisy voltages $\{y_t\}$, whereas in (d) we show samples from the posterior given $\{y_t\}$; the fact that the two sets of traces are similar is a useful check that the model fits have converged. (f) Voltages sampled from the rSLDS prior, corresponding to traces shown in (e). Note that the simple three-state two-dimensional rSLDS is able to learn to produce reasonably accurate spike shapes and firing rates.

Fig. 2(d). Moreover, simulating from the learned dynamics yields realistic trajectories in both latent space (e) and in voltage space, as shown in Fig 2(f).

Figure 3 compares the rSLDS to a simpler baseline method. Past work in voltage smoothing has utilized the Kalman smoother, based on an assumption of approximately linear dynamics in the subthreshold regime [14]. Notably, the Kalman smoother is a special case of the rSLDS with $K = 1$ discrete state. We compare performance over a range of noise levels and find that the rSLDS significantly outperforms the standard Kalman smoother, due largely to the fact that the former can adapt to the drastically different smoothness of the voltage signal in the spiking versus the subthreshold regimes.

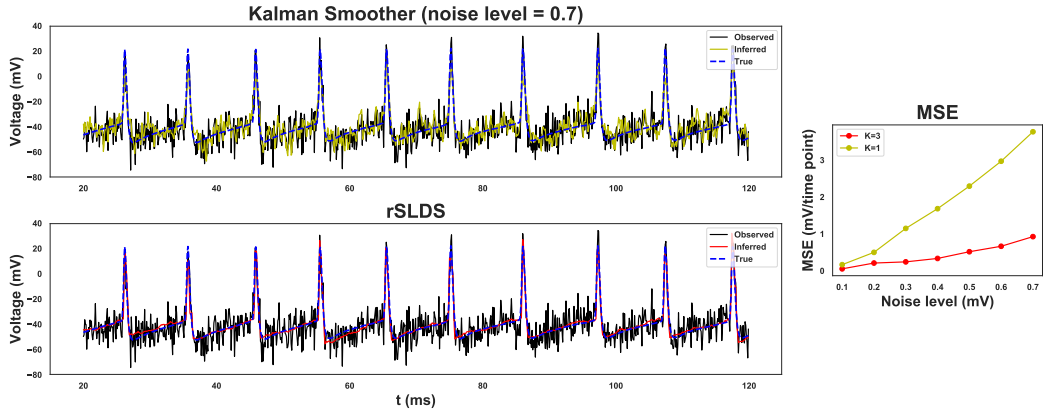

**Figure 3:** *Comparing the Kalman smoother against the rSLDS.* Left: example noisy data (constructed by adding noise to real voltage data) and output of the Kalman smoother (top) and rSLDS (bottom). Right: summary of MSE as a function of observation noise variance. The rSLDS outperforms the Kalman smoother, because the latter is a linear filter with a single homogeneous smoothness level, and therefore it must either underfit spikes or overfit subthreshold voltage (or both), whereas the rSLDS can enforce different levels of smoothness in different dynamical regimes (e.g., spiking versus non-spiking).

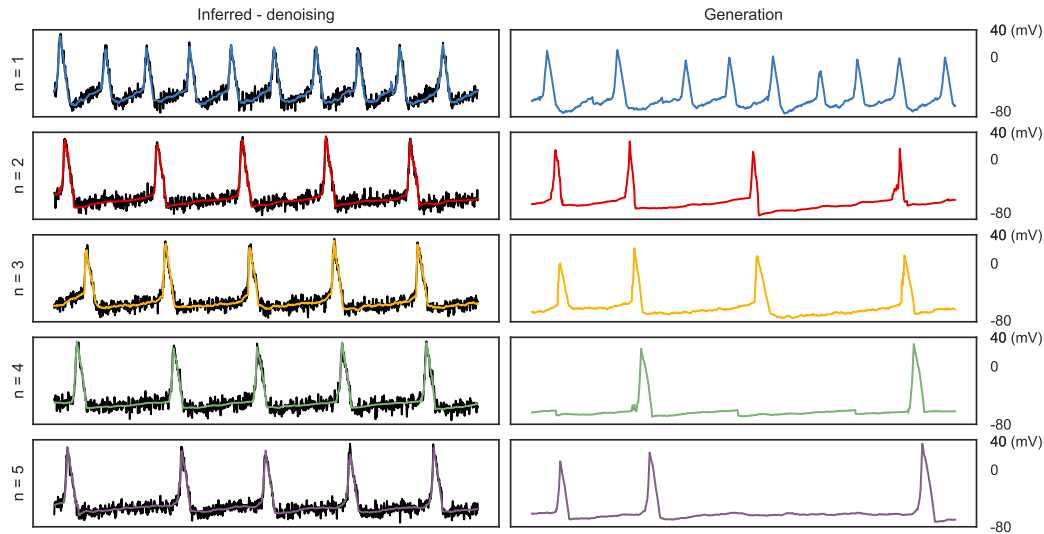

**Figure 4:** *Voltage smoothing in noisy recordings in a simulated dendritic branch.* The branch consists of five compartments connected in a chain. *Left:* The observed voltage (black line) and the inferred voltage (colored line) are shown for each compartment. *Right:* A sample from the learned multi-compartment rSLDS. The generated voltage traces show that the model has learned to reproduce the nonlinear dynamics of multi-compartment models, including the interactions between compartments that propagate spikes down the dendritic branch.

## 4.2 Spatiotemporal denoising with simulated data on real morphologies

The single compartment studies afford us ground truth electrical recordings, but obtaining simultaneous electrical recordings from a multiple compartments of a single cell is a significant technical challenge. Optical recordings of fluorescent voltage indicators offer multiple compartments, but at the cost of lower temporal resolution and significantly higher noise. To test our method's ability to denoise multi-compartment voltage traces, we simulate voltage traces using the simple HH model in lieu of ground truth electrical recordings[3]. We start by simulating a single dendritic branch and then move to a full dendritic tree, using real neuron morphologies.

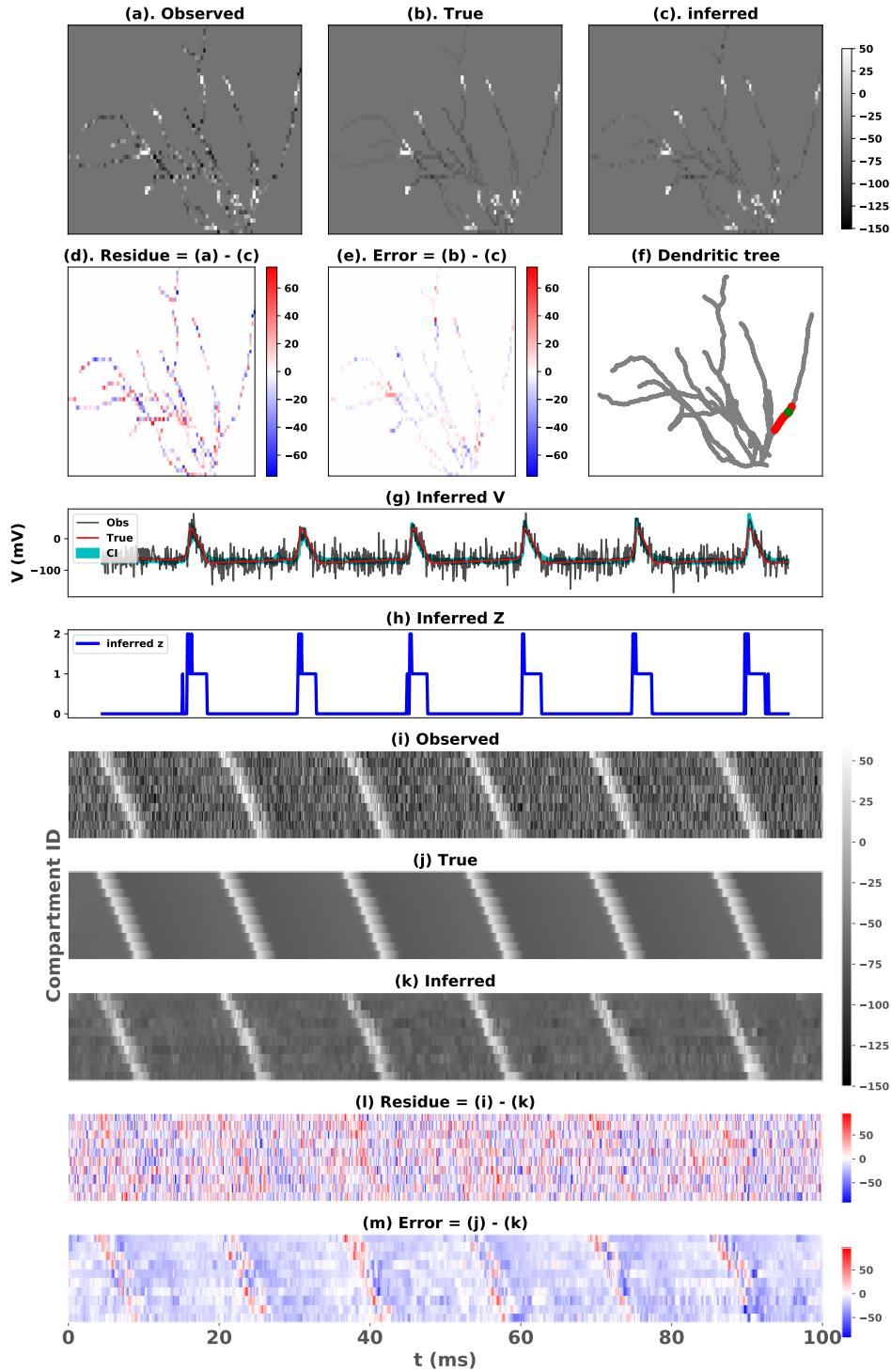

**Figure 5:** *Spatiotemporal denoising on a large simulated dendritic tree.* The 3D morphology of the tree is taken from a real cell in the Allen institute database. **(a)** Simulated voltage with additive Gaussian noise (25mV std.) on the dendritic tree. **(b)** True simulated voltage without noise. **(c)** Denoised estimate of voltage with the multi-compartment rSLDS; note the close match with (b). Colorbar shared between (a), (b), and (c). **(d)** Residual equals observed (a) minus inferred (c). **(e)** Error equals (b) minus (c). **(f)** Cartoon figure indicates the single compartment (green dot) and a segment of dendritic tree (red branch) shown in the following panels. **(g)** Noisy observation (black) and posterior credible interval (CI) of inferred voltage (cyan), and true voltage (red) corresponding to dot in (f). **(h)** The inferred discrete state of the compartment (0: subthreshold; 1: spike fall; 2: spike rise). **(i, j, k)** spatiotemporal representation of the noisy, true, and denoised voltage propagating up the branch shown in (f). **(l, m)** are the residual and error computed from (i, j, k). See Video link for full details.

**Denoising a dendritic branch.**

We model a single dendritic branch as a five-compartment chain. We simulate membrane potential according to the classical HH model with voltage-gated sodium and potassium channels, a leak current, and currents from neighboring compartments in the chain. We inject a constant 35mA current into the first compartment, and induced spikes propagate to downstream compartments according to the cable equation. However, we have tuned the inter-compartment conductances $g_{nn'}$ such that only every other spike propagates — a single spike in compartment 1 cannot depolarize compartment 2 enough to reach the spike threshold, but two spikes can. Moreover, we corrupt the voltage traces with additive Gaussian noise with standard deviation of 8mV. We test the multi-compartment rSLDS's ability to mimic the nonlinear dynamics of the true generative process and smooth the noisy voltage observations.

Figure 4 shows the five compartments in the chain. In the left column we show the observed voltage traces (black) and the smoothed voltage (color) traces. Again, the model does an excellent job denoising the data, and is also able to learn a generative model of the rich spatiotemporal dynamics. In particular, the model does not simply learn separate models for each compartment — it also learns interactions between compartments. This is evident in the traces that are generated by the model, as shown in the right column of Fig. 4. The first compartment shows a high firing rate, but only approximately every other spike propagates to downstream compartments, as in the real data. The generation is not perfect: some spikes fail to propagate (e.g. the third spike in compartment 3 does not propagate to compartment 4), and we see some spurious discrete state transitions (data not shown). Nevertheless, these generated samples indicate that the learned dynamical system captures the gross structure of multi-compartmental membrane dynamics. An accurate generative model offers a strong prior for smoothing spatiotemporal noisy voltage traces given by optical recordings.

**Denoising a full dendritic tree.**

We have shown that the rSLDS can learn the dynamics of single compartments and dendritic branches (i.e. multi-compartment chains), but can these methods scale to full dendritic trees? We test these models on real three-dimensional dendritic tree [27, cell id=464212183]. (Note that this morphological data only includes the three-dimensional shape, not any voltage recordings.) Of the 2620 compartments in the original tree, we retain approximately every fifth compartment to create a tree with 519 compartments (Fig. 5f). We simulate the HH model on this tree to obtain 100ms of data at 10kHz temporal resolution. As above, we add Gaussian white noise (uncorrelated in space and time; standard deviation 25mV).

Fig. 5a shows the observed voltage across all spatial compartments for a single time point, and panel (b) shows the true underlying voltage. Supplementary Video 1 shows the voltage propagating in time through the tree. Panel (c) shows the voltage inferred by the multi-compartment rSLDS. The residual in panel (d) and the error in (e) show the difference between the inferred voltage and the observed and true voltage, respectively. There is a slight spatial correlation in the errors — specifically, the inferred voltage tends to slightly underestimate spike amplitude and overestimate the voltage during recovery — but the errors are generally small. Panel (g) shows the temporal estimates for the single compartment indicated by the green dot in (f). The posterior credible interval captures the true voltage, despite the high level of noise. Each spike corresponds to a canonical discrete state sequence, as in shown in (h). By transitioning between these discrete states, the piecewise linear dynamics aid the model-based denoising. (As in Figure 3, we find that the $K = 1$ Kalman smoother tends to undersmooth the data here; results not shown.) Finally, panels (i-k) show both spatial and temporal dynamics of voltage for the dendritic branch highlighted in red in (f). We see the spikes propagating along the compartments in observed, true, and inferred voltage. Again, the residuals (l) and errors (m) show small spatiotemporal correlations, but overall good recovery of the voltage from the noisy observations.

For comparison, previous particle filtering-based approaches to voltage smoothing [15] would need to infer the trajectories of $\approx 2000$ state variables ($N = 500$ compartments with $D = 4$ dimensions each) in the simplest HH model of this data, and this model would not be able to adapt to modest changes in the dynamical parameters of the channels in each compartment. Scaling such methods to this number of state variables is a serious challenge, but the recurrent switching linear dynamical system approach makes this problem tractable.

# 5 Conclusion

The advent of new voltage imaging methods presents exciting opportunities to study computation in dendritic trees. We have developed new methods to smooth and denoise optical voltage traces in order to realize this potential. These methods incorporate biophysical knowledge in the form of constraints on the form of inter-compartmental dynamics, while allowing for effectively nonparametric learning of nonlinear intra-compartmental dynamics. We have illustrated the potential of these methods using semi-synthetic data based on real electrophysiological recordings of membrane potential and actual neuron morphologies. The results show promising progress toward the critical goal of denoising noisy spatiotemporal voltage measurements.

**Acknowledgements**

We gratefully acknowledge support from the ONR and Simons Foundation, and from NIH grants 1U01NS103489-01, 1U19NS113201, and EB22913.

## Footnotes

[2]In principle, all compartments' continuous latent states and voltages could be updated at once, given that they are conditionally jointly Gaussian; however, the computational cost would naively scale as $O(T(ND)^3)$, based on a Kalman forward-backward sweep. An important direction for future work would be to adapt the fast approximations proposed in [14] to implement this joint update more efficiently.

[3]Of course more detailed realistic simulations with multiple nonlinear channel types are possible; we leave this for future work. As we will see below, the simple HH model already provides a rich and interesting testbed simulated dataset for the methods presented here.

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
