[Reviews · NeurIPS 2019]

Reviewer 1



This paper proposes a novel application of recurrent switching linear dynamical systems to denoising dendritic voltages based on the cable equation. The model is able to approximate nonlinear voltage dynamics by switching between linear dynamics matrices as determined by a time-varying discrete latent variable. Each discrete state corresponds to a different stage of action potential generation: resting state, depolarisation, and hyperpolarisation. The approach outperforms the more standard Kalman smoother, because the latter fails to account for the fact that different stages of action potential generation require different levels of smoothness to be enforced. The modelling is high quality, the writing is clear, and the problem is well motivated. Details on the inference are rather lacking, although this is likely due to them be described in sufficient detail in the referenced articles. Major comments: - It becomes clear how the discrete latent states should be interpreted later on, but when the model is introduced it is difficult to understand what role they play. The initial description of the model (page 3) would benefit from a suggestion of how to think about discrete states in this context. Similarly, the continuous latent variables x_t^n are introduced on page 3, but it was never totally clear how the different dimensions should be interpreted (biophysical or otherwise). - The selection of K = 3 leads to a natural biophysical interpretation of the discrete latent states. Was this the intention a priori? What happens as K increases? - The manuscript title and abstract push scalability as a major theme, yet this only seems to be addressed in a brief paragraph (page 8) without much quantitative evidence. Minor comments: - The panel labels have inconsistent styles. Some are just the label with a period, some have parentheses with a period, some have parentheses without a period. - Title of panel 5c: Inferrd -> Inferred. Update: The author response addressed my concerns. I maintain my vote for accepting this paper.

Reviewer 2



The authors describe a method of denoising of optical membrane voltage recordings using the rSLDS as a proxy for membrane currents in the standard membrane conductance model. They leverage inference techniques pioneered in the rSLDS literature and demonstrate that their method is both reasonable to the “eyeball test” but also quantitatively out-performs the state of the art. The paper is exceptionally well written, easy to follow, technically sound, and the experiments were well designed to progressively illustrate the performance of the method. The model is novel and makes good use of both statistical and biophysical models. Clearly, it would have been preferable to demonstrate their method on real data as well, but this should not be disqualifying. I should note that the reproducibility checklist answers indicated that the authors have downloadable code but I was not able to identify where that code was.

Reviewer 3



This study develops a Bayesian approach to denoise dendritic voltage traces in the context of large dendritic trees. Below, a review according to relevant criteria. Originality: The novelty of the study is two fold: (1) it resides in the usage of the recently developed model class of recurrent switching linear dynamical system (rSLDS) to do (temporally-)locally linear approximations of the non-linear interactions between membrane channels and voltage in dendritic trees; (2) such approximation allows for the usage of the machinery of rSLDS to perform inference of the latent voltage traces, given noisy voltage traces in a large number of compartments of a dendritic tree. The manuscript makes it clear how it differs from previous work (in particular, regarding inference of dendritic voltage traces), and related work on inference with rSLDS is appropriately cited. Quality: The methodology, results and analyses are insightful and technically sound. Clarity: The manuscript is clearly written, and provides enough information for an expert reader to understand all the steps to reproduce the results. Some minor comments to improve clarity and some typos: -the manuscript should discuss how the levels of noise added to the voltage data in the numerical experiments compare to the noise observed in real voltage imaging data; -in Figure 3, it would be good to have a quantitative comparison between the Kalman smoother and rSLDS; -throughout the paper, "data" is used as a singular word, although it is a plural word, so this should be corrected; -line 173 "the model does an excellent job denoising the data". Instead of "excellent", it would be good to have a more quantitative/objective statement about the quality of denoising; -lines 196-198: "There is a slight spatial correlation in the errors...". Although perhaps trivial, a short discussion of the reasons for this and solutions would be desirable. Significance: The methodology and results presented are of interest to both the computational as well as the experimental neuroscience community and will inspire applications to real data, and further method development. Indeed, the approach and respective results presented are a substantial improvement compared to previous approaches, in particular regarding the expressive power of the latent voltage dynamics and the dimensionality of the voltage signals (number of dendrites in the dendritic tree), and constitute a solid step towards applications in real datasets. -- After the rebuttal -- The authors' response addressed satisfactorily all my comments, and I therefore maintain my support for the acceptance of this paper.

[Author Response · NeurIPS 2019]

1 We thank the referees for their time and the kind reviews. Brief responses follow.

**Model selection**

We explored different values of the parameter $K$. The value $K = 3$ achieves robust performance in both training and test data, and is interpretable biophysically, and so we focused our attention on $K = 3$ here. We found that smaller values of $K$ led to worse performance, and higher values of $K$ could lead to unstable learning. We plan to present further details of these analyses in an appendix to the final paper.

**Real data**

We are currently applying these methods to real voltage imaging data; preliminary results are encouraging. This work will most likely be described in a separate paper. We also emphasize that many experiments in our submission do use real voltage traces (corrupted with artificial noise, Figures 2+3) and neuron morphologies (with simulated voltage traces, Figure 5), allowing us to assess recovery of ground truth voltage via semi-synthetic data.

**Quantitative comparison between the Kalman smoother and rSLDS**

We have included this comparison in the right panel of Figure 3; we will clarify this point in the revised text.

**Interpretation of $X^{(n)}$**

These are auxiliary continuous latent variables that the rSLDS uses to model the voltage dynamics. Intuitively, the second dimension helps determine whether the voltage is rising or falling, which is an important signal for the discrete state transition probabilities in the rSLDS. We will clarify this in the revised text.

**Other issues**

We will upload the code (as suggested by R2) and fix typos; thank you for pointing these out. We will add material in the appendix clarifying the scalability of the method (as R1 suggested). We will introduce the biophysical meaning of the discrete latent variable earlier in the paper (as R1 suggested) and provide discussions about "slight spatial correlation errors" (as R3 suggested). Thanks again for these helpful suggestions.

[Meta-Review · NeurIPS 2019]

The article considers a novel application of the recurrent switching linear dynamical system model to denoising dendritic voltages. The paper is very well written, the problem well motivated with compelling experiments. A very solid application paper overall. [This meta-review was reviewed and revised by the Program Chairs]